# Project OPUS: Development and evaluation of an electronic platform for pain management education of medical undergraduates in resource-limited settings

Tonia C. Onyeka[1,2], Nneka Iloanusi[2,3], Eve Namisango[4], Justus U. Onu[2,5], Kehinde S. Okunade[6], Alhassan Datti Mohammed[7], Muktar A. Gadanya[8], Abubakar U. Nagoma[7], Samuel Ojiakor[9], Chukwudi Ilo[10], Okey Okuzu[11], Chinelo Oduche[11], Ngozi Ugwu[12], Matthew J. Allsop[13]*

1 Department of Anaesthesia/Pain & Palliative Care Unit, College of Medicine, University of Nigeria Ituku-Ozalla Campus, Enugu, Enugu State, Nigeria, 2 EPAC Research Team, College of Medicine, University of Nigeria Ituku-Ozalla Campus, Enugu, Enugu State, Nigeria, 3 Department of Radiation Medicine, College of Medicine, University of Nigeria Ituku-Ozalla Campus, Enugu, Enugu State, Nigeria, 4 African Palliative Care Association, Kampala, Uganda, 5 Department of Mental Health, Faculty of Medicine, Nnamdi Azikiwe University, Awka, Anambra State, Nigeria, 6 Oncology and Pathological Studies (OPS) Unit, Department of Obstetrics & Gynaecology, College of Medicine, University of Lagos/ Lagos University Teaching Hospital, Lagos, Lagos State, Nigeria, 7 Department of Anaesthesia, College of Health Sciences, Bayero University/ Aminu Kano Teaching Hospital, Kano, Kano State, Nigeria, 8 Department of Community Medicine, Bayero University/Aminu Kano Teaching Hospital, Kano, Kano State, Nigeria, 9 Department of Anaesthesia, College of Medicine, Nnamdi Azikiwe University, Awka, Anambra State, Nigeria, 10 College of Medicine, Enugu State University of Science and Technology, Enugu, Enugu State, Nigeria, 11 InStrat Global Health Solutions, Abuja, Nigeria, 12 Department of Haematology, Alex Ekwueme Federal University Teaching Hospital, Abakaliki, Ebonyi State, Nigeria, 13 Academic Unit of Palliative Care, Leeds Institute of Health Sciences, University of Leeds, Leeds, United Kingdom

* m.j.allsop@leeds.ac.uk

**Data Availability Statement:** All relevant data are within the manuscript and its Supporting Information files.

## Abstract

### Introduction

Pain is a very frequent symptom that is reported by patients when they present to health professionals but remains undertreated or untreated, particularly in low-resource settings including Nigeria. Lack of training in pain management remains the most significant obstacle to pain treatment alongside an inadequate emphasis on pain education in undergraduate medical curricula, negatively impacting on subsequent care of patients. This study aimed to determine the effect of a 12-week structured e-Learning course on the knowledge of pain management among Nigerian undergraduate medical students.

### Methods

Prospective, multisite, pre-post study conducted across five medical colleges in Nigeria. Structured modules covering aspects of pain management were delivered on an e-Learning platform. Pre- and post-test self-assessments were carried out in the 12-week duration of the study. User experience questionnaires and qualitative interviews were conducted via

**Funding:** The e-Learning platform development was funded by the International Association for the Study of Pain (IASP) through the 2019 IASP Developing Countries Project: Initiative for Improving Pain Education Grant, a scientific grant awarded to T.C. Onyeka, M. Allsop, N. Iloanusi and E. Namisango. IASP funding supported the development of the e-Learning platform. Institutional support was provided by the College of Medicine, University of Nigeria (Enugu, Nigeria) and the African Palliative Care Association (Kampala, Uganda) to support staff costs to undertake research and evaluation activities. The funders provided support in the form of salaries for authors [TO, EN], but did not have any additional role in the study design, data collection and analysis, decision to publish, or preparation of the manuscript. The commercial partner for this study, InStrat Global Health Solutions [OO, CO], did not provide funding to support this project. InStrat Global Health Solutions did not contribute to the study design, analysis or decision to publish. The platform developed by InStrat Global Health Solutions supported the collection of study data entered by study participants and the company provided technical input into the manuscript, supporting the description of the functionality of the eLearning platform. The specific roles of these authors are articulated in the 'author contributions' section.

**Competing interests:** Okey Okuzu is Founder and CEO, and Chinelo Oduche a Project Manager, for InStrat Global Health Solutions, the company providing the software on which the e-Learning platform was developed in collaboration with the research team. This commercial affiliation does not alter our adherence to PLOS ONE policies on sharing data and materials. All other authors declare no conflict of interest.

instant messaging to evaluate user experiences of the platform. User experience data was analysed using the UEQ Data Analysis Tool and Framework Analysis.

## Results

A total of 216 of 659 eligible students completed all sections of the e-Learning course. Participant mean age was 23.52 years, with a slight female predominance (55.3%). Across all participants, an increase in median pre- and post-test scores occurred, from 40 to 60 (Z = 11.3, p<0.001, effect size = 1.3), suggestive of increased knowledge acquisition relating to pain management. Participants suggested e-Learning is a valuable approach to delivering pain education alongside identifying factors to address in future iterations.

## Conclusion

e-Learning approaches to pain management education can enhance traditional learning methods and may increase students' knowledge. Future iterations of e-Learning approaches will need to consider facilitating the download of data and content for the platform to increase user uptake and engagement. The platform was piloted as an optional adjunct to existing curricula. Future efforts to advocate and support integration of e-Learning for pain education should be two-fold; both to include pain education in the curricula of medical colleges across Nigeria and the use of e-Learning approaches to enhance teaching where feasible.

## Introduction

Pain is a very frequent symptom that is reported by patients when they present to health professionals [1]. It is defined by the International Association for the Study of Pain as "an unpleasant sensory and emotional experience associated with, or resembling that associated with, actual or potential tissue damage" [2]. For a multitude of reasons, it remains under-treated or untreated in low-middle income countries (LMICs) like Nigeria [3]. Common causes include poor knowledge and attitudes about pain relief, limited pain treatment facilities, restrictive government policies, high costs, socio-cultural challenges and problems with access and use of analgesic medications, especially opioids [4, 5]. This is despite research demonstrating improvement in healthcare providers knowledge of pain and its management usually results in a reduction in patients' pain experience [6].

Lack of training in pain management remains the most significant and fundamental obstacle to effective pain management as an inadequate emphasis on pain education for medical undergraduates may eventually reflect in poor patient care practices following graduation [7–9]. Medical students often begin their education in Colleges of Medicine with little or no knowledge of pain and that situation, coupled with very few hours of pain training, results in their lack of confidence in assessing adult and paediatric patients with pain [10]. It is thought that existing prevalent negative attitudes of physicians toward patients with, for example, chronic non-cancer pain begins early in medical school [11]. In many developed countries, knowledge-based learning of pain is absent of emotional development and reflective capacity, hindering the ability of medical students to develop empathy [11].

Integration of pain education in undergraduate medical curricula varies across countries at all levels of development [12]. In Europe, 7% of medical schools lack any pain component in their curriculum [13]. However, in Canada around 92% of Canadian medical schools and 80%

of medical schools in the United States have mandatory pain management content in the medical undergraduate curricula [14]. In the context of developing countries the gap in provision is starker. For example, in Nigeria, only three of over forty medical schools offer some form of pain management training. Lectures in pain management are not stand-alone courses, but tutorials embedded within the Anaesthesia clerkship; the latter having an average duration of three weeks and thus potentially leaving the medical students insufficiently exposed to pain management education. While there is a huge need to adapt existing medical school curricula to accommodate a new subspecialty curriculum around pain management, there is limited leadership for this at present in Nigeria. The medical curriculum in its present form is overloaded and would require foregoing existing elements of teaching to create capacity to include pain management training [15]. Therefore, in the near term, the authors consider the e-Learning approach to be a possible viable adjunct to support access to pain management education alongside the existing medical school curriculum.

An ever-evolving approach to delivery of training for those delivering healthcare is through utilising technology, often referred to as e-Learning [16]. e-Learning approaches have several synonyms which include computer-based learning, online learning, distance learning and web-based learning [17]. It has been defined as, 'a learning process that involves the connection of digitally-conveyed content, system-based administrations and mentoring bolster' [18]. Choosing to explore e-Learning approaches enables many advantages, including flexibility to learners' needs and time commitment, opportunities for standardization of content, and built-in pre- and post-test evaluation capabilities [19, 20]. In addition, e-Learning provides content that may not be accommodated by routine classroom teaching while at the same time allowing students to learn at their own pace. There is good evidence that such online pain educational resources are effective at improving learner knowledge, and this has been demonstrated across a number of high-income country settings, including Canada, Finland, Germany, Italy and the USA [21, 22]. This is aligned with the general increase in the provision of online distance education worldwide [23]. However, e-Learning approaches to pain management have not been explored in the context of sub-Saharan Africa in countries such as Nigeria, where poor and costly internet access and irregular power supply may pose barriers. For example, no public Nigerian university, to the best of our knowledge, offered any online courses prior to the COVID-19 pandemic, forcing schools and higher education institutions to temporarily shut down. Online services in universities are often utilized for payment of school fees, staff salaries and to enable prospective students to attempt entrance exams or allow successful students' register, but not as a mode for delivering teaching and learning [24]. This study sought to address this gap through the development and evaluation of an e-Learning approach to pain education, focusing specifically on pain knowledge, for medical undergraduates in Nigeria. It was hypothesised that delivery of pain management lectures using an e-Learning platform would lead to an increase in the pain knowledge of medical undergraduate students.

## Methods

### Context of system development

This work was undertaken as part of seed funding awarded by the International Association for the Study of Pain. The grant call sought to support initiatives for improving pain education and practice in developing countries. The remit of proposals was for one-year projects that were ready to begin within four months. The decision to develop a proposal focused on an online platform brought together an emerging collaboration on digital technologies between two of the manuscript authors (TO and MA), a previous collaboration between an IT company (InStrat Global Health Solutions (InStratGHS)) and a member of the research team (MA), and

an established research partnership on the development of digital technologies for pain and palliative care research across sub-Saharan Africa (EN and MA).

## Design

We adopted a research-led development process of an e-Learning platform followed by a pre-post study design to evaluate its use. The platform was designed for 5[th] year undergraduate medical students as participants (N = 659) and then deployed across five accredited Medical Colleges in Nigeria sites over a 12-week implementation period. Participation of students was voluntary, informed consent was obtained from each participant. The investigators first obtained ethical clearance from a local human investigations committee, the University of Nigeria College of Medicine Research Ethics Committee (COMREC; Protocol No: 079/09/2019). Further consent was obtained from local ethics committees for each of the other participating sites (Health Research Ethics Committee, College of Medicine, University of Lagos. Approval no: CMUL/HREC/09/19/616; Health Research Ethics Committee, Ministry of Health, Kano State. Reference no: MOH/Off/797/T.1/1818; Health Research Ethics Committee, College of Medicine, Enugu State University of Science and Technology. Reference no: ESUTHP/CMAC/RA/034/Vol1/218). The 12-week long program which was self-paced/asynchronous, was intended to help participants learn to define pain and identify types/forms of pain, understand the ethics of pain, learn the various pain assessment methods and be able to treat pain, particularly in special conditions.

## Overview of platform development

The project team which was a multidisciplinary team led by a pain specialist (TO), a global palliative care researcher (MA), a global health researcher and person-centred care service development fellow of the African Palliative Care Association (EN) and InStratGHS, a mobile health technology company. Platform development followed the disciplined agile delivery (DAD) methodology, which is a formal structure used by software developers to guide health information technology system development from the initiation of ideas through implementation and eventual retirement [25]. The DAD methodology shares principles of approaches often used to develop interventions in health research, such as user-centred design [26] and participatory design [27], where the stakeholder, or end user of a technology or product (in this case, the medical student), is central to its design and development. Working within the DAD framework provided a clear development process for the system developers. It also provided clear time points for the research team, highlighting timepoints for curriculum development, initial assessment of prototypes of the platform by the research team, and agreement on content for pre- and post-test items included as part of the platform.

The DAD framework plans system development over four phases: inception, elaboration and construction, and transition. The inception phase of the project began with the team generating a working technical specification document, which outlined the planned components and functions that were initially deemed necessary for the online platform for pain education. During the subsequent elaboration and construction phases, the team developed a curriculum and supporting materials to form the content of the platform. The transition phase involved the implementation and evaluation of the platform across five accredited Medical Colleges in Nigeria. The research team adopted a mixed-methods approach, combining user engagement activities (e.g. needs assessment) to guide the development of the curriculum delivered via the online platform, quantitative assessment of changes resulting from use of the platform pre- and post-implementation, and then used qualitative interviews to understand the experience of students who used the system and the facilitators and barriers of use of the system. We

outline the methodology used during the platform development and evaluation below, aligned with the stages of the DAD framework.

## The inception phase

This phase began by the research team undertaking a review of materials to shape the scope of the curriculum. This involved systematic searching of literature across Medline, Embase and PsycInfo databases using keywords such as pain education, undergraduate, medical education, and low-resource setting. Information on existing undergraduate pain curricula was sought from eight potential implementation sites of the platform. Details sought on pain education included the form of pain education provision, hours spent teaching pain, pain topics covered as well as methods of teaching and assessment. Alongside mapping out existing pain education, we received input from experts on the availability of materials to inform the development of the platform. This included consultations with the African Palliative Care Association which is a pan-African non-governmental organization working to promote and support the integration of palliative care, including pain and symptom management, into health systems across Africa, and the identification of open access materials from several pain websites that could guide the development of platform content.

Alongside consultation on the content of the platform, discussions took place between the research team and InStratGHS about the functionality of their existing platforms. Relying on the requirements of the project early in the development enabled InStratGHS to better understand the needs of the project and identify a suitable solution for deploying the pain education curriculum. The company selected VTR Mobile, the proprietary mobile learning platform of InStratGHS, for the project, which is a self-paced learning platform that can be used both online and offline on an Android smart phone, tablet computer, or laptop/desktop computer. It features a mobile application which allows users to view training contents offline where there is no internet connectivity or to save data. VTR Mobile supports multi-media training content and its test taking features allowed results to be generated in real time while its back-end provided real-time user participation statistics and test results.

## The construction and elaboration phases

This phase involved the creation of the pain module content and learning materials by the project team. We utilised the e-learning systems' theoretical framework to guide the design of the platform, comprising three components: people, technologies, and services [28]. The feasibility, practicalities and relevance of the different components were discussed. From consultation with sites during the Inception phase, feedback suggested that it would not be possible to incorporate the online platform into the existing undergraduate curriculum. The platform would have to supplement existing teaching activities and provide a standalone resource for students to access. Furthermore, the fifth year of study was chosen as the target group for the platform, being mid-way through students' clinical studies with adequate familiarity with clinical entities that require pain management. In Nigeria, the 4th year is the beginning of the clinical training during which time students are introduced to medicine and surgery. By the 5th year, the curriculum includes paediatrics, obstetrics and gynaecology as well as community medicine. This is followed by clinical training in medicine and surgery in the 6th and final year of study. The majority of the project team are experienced teaching staff delivering training to medical undergraduates in the regions of Nigeria included in the study so were able to ensure the content and curriculum were developed appropriately for 5[th] year medical students.

Discussions across the research team, technology developers and university sites guided both the technologies and services components of the platform. The pedagogical models and

instructional strategies were informed by the supplementary approach required for the platform. An open learning model was adopted, enabling flexibility and inclusivity of access to the platform content. A combination of instructional strategies was incorporated into the presentation of content, including contextualising instruction, activating and assessing learner outcomes and presenting and cueing content. Key resources used for developing the content identified during the inception phase included the detailed International Association for the Study of Pain (IASP) pain curriculum for medical graduates which was first developed in 1988 [29]. Inputs were also made from Beating Pain [30], a pocket text book on pain designed for self-directed reading that teaches the 'total pain' concept, the multidisciplinary approach to pain management as well as care of all aspects of pain developed by the African Palliative Care Association.

The review of existing resources was led by the research team (TO, NI) with expertise in the teaching and clinical practice of pain management in Nigeria. The research team developed a provisional pain education curriculum for undergraduate medical students informed by activities conducted during the inception phase, including feasibility of delivery within existing curricula (from consultation with potential implementation sites) and availability of resources from literature searching. The research team determined that, in the absence of specific pain education curricula, the structure should align where possible with the IASP pain curriculum for medical graduates [29]. An initial curriculum was developed, aligned to the IASP pain curriculum for medical graduates, which was supplemented by content specific to the sub-Saharan Africa from Beating Pain [30]. Examples of supplementary content included, for example, factors that lead to women in Arica being more likely to suffer pain than men, context-specific pain and symptom outcome measures, and pharmacological management. The initial curriculum was shared with a panel of four experts in the field of pain management, palliative care, paediatrics and geriatrics drawn from different parts of the African continent, who were independent of the project team. Comments were requested from the panel of experts on framing the content for undergraduate medical education, outlining the essential content for inclusion, and the optimal structure of modules. A first round of feedback was received on the overall curriculum, its content and overview of modules. A second round involved detailed feedback on the wording of module content alongside accompanying video and media materials. The final curriculum for pain education to be delivered using the online platform was agreed following two iterations of feedback from the expert panel. An overview of the curriculum is provided in Table 1 below and comprised six modules.

Following the development of the curriculum, supporting written and video materials were developed by the research team and shared with InStratGHS. A prototype was developed and full access provided to the research team for review. Following feedback, InStratGHS provided a second and final iteration of the online platform for implementation across sites in the transition phase. Technical specifications of the components of the final online version of the platform is outlined in Table 2 below.

## The transition phase

In this phase, the platform was deployed across multiple sites (N = 5). Eligible sites were medical schools with full accreditation status from the National Universities Commission (NUC) and the Medical and Dental Council of Nigeria (MDCN), whose Provosts or Deans had given consent to the study and who had obtained ethical clearance for the training from their respective institutions. Each site was assigned a site coordinator from faculty staff to provide local oversight and coordination of the project (JO, KO, DM, MG, AN, SO, CI, NU). The site coordinator also served as an intermediary between the class and the project team. Members of the

**Table 1. Overview of the pain education curriculum of the online platform.**

| Module title | Overview of Module |
|---|---|
| Module 1; Multidimensional nature of pain | • Definition of pain and classification of pain |
| | • Consequences of untreated pain. |
| Module 2: Neuroanatomy, Neurophysiology and Neuropharmacology of Pain | • The neuroanatomy and neurophysiology of pain |
| | • Classification of analgesics and their pharmacological targets |
| Module 3: Psychology of pain | • Psychological aspects of pain management |
| | • Pain-related beliefs and illness behaviours of patients with chronic pain |
| | • Cultural differences in pain meanings and treatment approaches |
| Module 4: Pain assessment | • Barriers to pain management |
| | • Popular pain myths |
| | • History-taking and measurement of pain in children and adults |
| | • Pain tools |
| | • Pain documentation |
| | • Uses and value of pain apps |
| Module 5: Treatment of pain | • Forms of pain treatment |
| | • WHO analgesic ladder |
| | • Analgesic treatment principles |
| | • Non-pharmacologic treatment of pain |
| Module 6: Pain in Special situations | • Principles of pain control |
| | • Special pain situations: Pain emergency, Postoperative pain, Pain syndromes, Labour pain, Cancer pain |
| | • Substance abuse and pain management |

InStratGHS team provided technical support where required at each site. Participating sites did not commence the training simultaneously but commenced after ethical clearance was given. Each site coordinator and dedicated InStratGHS staff, together with the Principal Investigator (TO), held an initial face-to-face meeting and demonstration to introduce and familiarize the students with the rationale for the platform and provide an overview of the included modules. At each site, the online platform (see Fig 1) was presented to students as a supplementary resource, independent of their progression to the final year and independent of the students' continuous assessment scores. The training was launched 12 hours following a site demonstration at each study site after which participants received personalized login details (username and password) alongside details for accessing the platform via mobile device or laptop. For the duration of the study, the site coordinators and InStratGHS staff were in constant

**Table 2. Overview of the technical specification of the online platform.**

| Technical aspect | Details of online platform |
|---|---|
| File Format | Documents: PDFs; Videos: MP4 |
| Data storage location | Encrypted cloud server on Amazon Web Services (AWS) |
| Data export format | .txt or.csv files |
| Maximum File Size | 65MB |
| Total download requirements across all modules | 750MB |
| Required User registration information | First Name, Last Name |
| Testing Format | Multiple Choice Options |
| Pass Threshold | 80% |

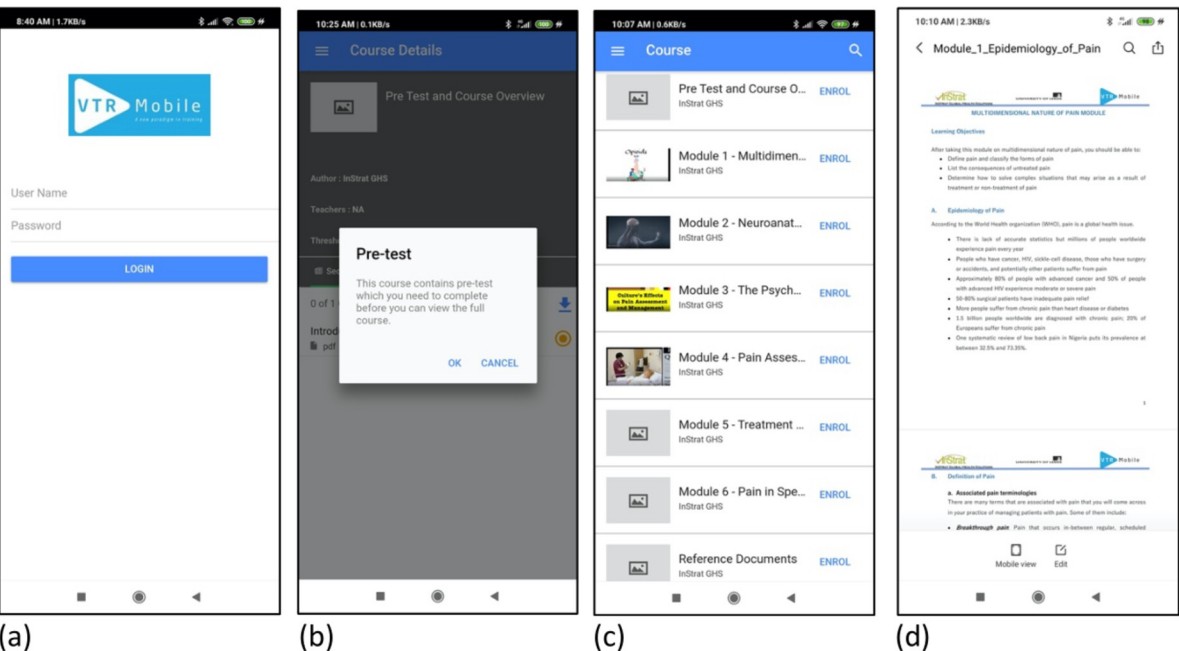

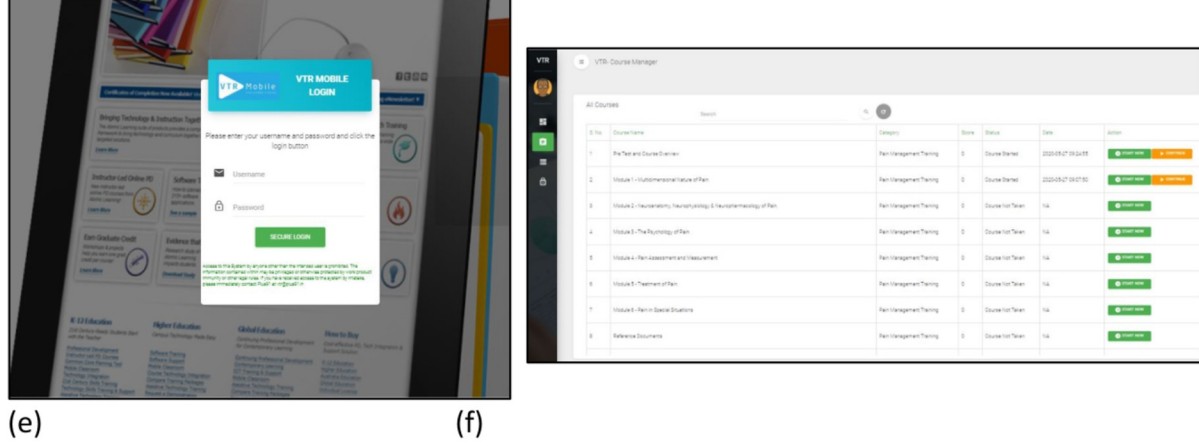

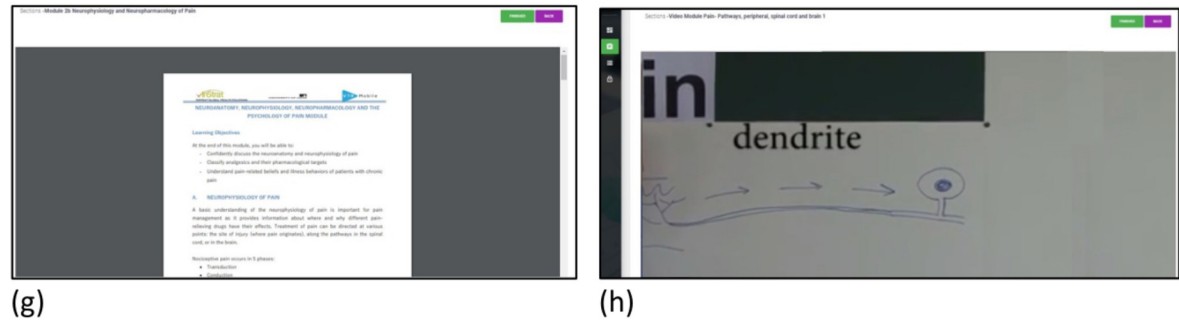

**Fig 1.** Screenshots of the interface for the VTR mobile app (a = login screen; b = pre-test notification; c = module list; d = example of written content) and the web application (e = login screen; f = module list; g = written content; h = video content).

communication with students, engaging in active discussions with them via the WhatsApp groups created to address any technical or other queries that arose.

The deployment of the e-Learning platform in this study adopted an asynchronous, self-learning approach, omitting a function that enables messaging between teachers and students. The platform functions included some interactive elements, such as the selection of the courses by platform users and options to view or skip any course, alongside the ability to take and retake the tests. The content, written content and videos, was presented as static material that students could access and view, but there were no interactive elements to the content itself. Each module comprised of an introductory video summarizing the module content, text-based materials such as written documents and slides, and videos of pain experts and pain procedures related to the module. Video content was in two forms; compulsory videos and optional videos. References to relevant reading materials and websites were also placed at the end of each module to encourage the participants to engage in further reading.

## Data collection

Prior to accessing the modules, each student was required to complete a pre-test survey to assess their baseline knowledge of pain and its management. The pre-test comprised of twenty single-correct option, multiple choice questions, each allotted a mark of 5 points with no bench-mark total score. Modules were then completed sequentially through the module numbers. At the end of each module, students were required to complete an end-of-module assessment, comprising of five multiple choice based on content from the module, with a pass mark of 80 percent set by consensus, (i.e. at least 4 correct questions out of 5) to qualify for progression to the next module. If the participant's test score on an end-of-module assessment was below 80, he/she had the option to repeat the quiz as many times as they deemed necessary to achieve a pass mark score of 80 and above in order to progress to the next module. On completion of all six modules, a post-test and the User Experience Questionnaire (UEQ) were completed once online and submitted. The post-test replicated the same items as those presented during the pre-test and both were used to assess changes in pain knowledge. Completion of the post-test after 12 weeks was a requirement for accessing/obtaining a certificate of completion and this was provided to students by the respective site coordinators. The certificate of completion was accredited and issued by the African Palliative Care Association.

The User Experience Questionnaire (UEQ) [31] contains 26-items (adjective pairs) and is designed to evaluate the user experience of a new product in situations where the product has no previous user experience evaluation. The questionnaire assesses a product for its pragmatic (task-oriented) and hedonic (non-task oriented) qualities. Each item is rated as a 7-point Likert scale and answers are scaled from -3 (fully agree with negative term) to +3 (fully agree with positive term) and 0 referring to a neutral answer. The 26 items align with six high-level scales linked to the attractiveness, perspicuity, efficiency, dependability, stimulation and novelty of a product [32]. The general impression the participants had about the Project OPUS training platform was measured by the attractiveness scale. Perspicuity refers to ease or difficulty encountered in using the platform while efficiency measured user interface look and dependability measured the participants' expectation of the online platform. The level of excitement and interest the participants had while using the platform was measured by the stimulation scale while the novelty scale measured innovativeness and creativity of the platform.

All data entered on the InStratGHS VTR Mobile platform was stored on its secure encrypted cloud server on Amazon Web Services (AWS). Access to this data is possible only with assigned Administrative login and passwords. Data from the pre-test, end of module assessments, post-test and UEQ as well as user engagement data (registration date, course

progress and test taken, test scores, and error logs) were all captured via the InStratGHS VTR Mobile platform and stored on AWS. An InStratGHS Administrator was responsible for maintaining the password-protected access to the reporting portal that was hosted on the secure web server. Error logs were generated for those students who required technical support. All study data were exported from the InStratGHS system into Microsoft Excel for ease of transfer to appropriate statistical software packages. The downloaded data was reviewed for quality assurance and securely shared with the research team for further review and analysis.

Following the completion of all modules, site coordinators at all sites contacted students to invite them to participate in an online group discussion on perceptions about e-Learning which was held utilizing WhatsApp. Two sites, BUK and ESUT, reflected sites that had relatively high levels and low levels of engagement on the online module respectively. Participants who had used the e-Learning and those that had not participated in the online learning were invited to participate. A topic guide was used to direct discussions, focusing on the reasons for uptake and non-use of the e-Learning platform, its role in supporting pain education, and for those who used the platform, experiences in their use of the platform.

## Data analysis

Data included pre-test and post-test scores, module test scores, user experience questionnaire data, and transcripts from semi-structured interviews. To describe the characteristics of students and different sites in the study population, relevant descriptive statistics (frequencies, means and standard deviations as appropriate) were produced. The normality of distribution of the data was assessed using the Shapiro-Wilk test. Comparison of the pre-and post-test scores for all participants was undertaken using the Wilcoxon sign-ranked test with effect size calculated at 95% confidence interval while the Kruskal-Wallis test used to compare pre- and post-test scores across the five participating sites.

User experience data was analyzed using the UEQ Data Analysis Tool® version 7, which is freely available from the UEQ homepage (http://www.ueq-online.org). The tool calculates the scale means and the mean and standard deviation per item. Values between -0.8 and 0.8 represent a more or less neutral evaluation of the corresponding scale, values > 0.8 represent a positive evaluation and values < -0.8 represent a negative evaluation. The tool groups the 26 items to create scores for 6 domains of attractiveness, perspicuity, efficiency, dependability, stimulation and novelty. Means scores were calculated for each domain. The tool also produces a benchmark graph comparing the product against a benchmark dataset of scores from $\approx$ 250 product evaluations using the UEQ including business applications, development tools and web shops and services [32]. As part of understanding user experience, technical reports were classified into higher-level categories of issues experienced with login, downloading content, video content and app installation, and the frequencies of occurrence calculated.

Following completion of the focus group interviews, data was extracted from the WhatsApp Messenger groups after download from site coordinator's phone and turned inserted into Microsoft Word documents to create interview transcripts. The analysis was led by a member of the team (MJA). Interview transcripts were analysed using Framework Analysis [33] to draw out key themes from the data. The Framework Analysis process involved five key stages: (1) Familiarisation—getting an overview of the issues raised during the interviews; (2) Identifying a thematic framework—making notes on the key issues discussed; (3) Indexing—applying the thematic framework to the data; (4) Charting—moving data from individual interviews and putting sections into the framework; (5) Mapping and interpretation—the researcher attempts to make sense of the data and interpret the key themes and issues discussed.

## Results

The cohort of Nigerian medical students eligible for this program was 659. Of the 659 invited participants, 326 commenced the training (see Fig 2), with a total of 219 students completing all sections of the e-Learning programme, providing a completion rate of 33.2%. The mean age of the students was 23.52 years, with a slight female predominance (55.3%). There was wide variation in levels of completion across sites from invitation to completion of all modules on the e-Learning platform, ranging from 4.7% to 74.8%.

### Pre- and post-test scores and level of change

The modules scores for both pre- and post-test were found not to be normally distributed. The median pre-test score of all participants was 40.0 (IQR = 20.0), increasing to 60.0 (IQR = 25.0) following completion of six modules on the e-Learning platform. A Wilcoxon signed-rank test outlined that differences between pre- and post-test scores were significant (Z = 11.3, p<0.001). For all sites, there was a marked improvement in the post-test as evidenced by statistically significant increases in median scores differences and related effect sizes (Table 3).

There was no statistically significant difference in the median of the pre-test scores across all the schools ($p$ = 0.26) (Table 4) suggesting a homogenous composition of baseline pain knowledge. There were however statistically significant differences in post-test score between the different sites, [$\chi2(4)$ = 10.7, p = 0.03]. In addition, the highest post-test scores across all sites occurred in Module 6 while the lowest post-test scores were seen in Module 1 (Table 5).

### User experience

In total, user experience data was obtained for 46 participants across the five sites. For those who completed the e-Learning programme and provided user experience data, the overall user experience rating was positive. In particular, the platform was rated positively for the attractiveness (mean score = 1.699) and stimulation domains (mean score = 1.750) (Fig 3).

The specific responses to individual items provided by participants are shown in Fig 4. Overall, across items, responses were positive for most participants. Items where the e-Learning platform received the most negative responses for over half of respondents was where respondents reported the system as being "slow" and "unpredictable".

When plotted against benchmarking data (presented in Fig 5) from comparative online tools and web applications, the e-Learning platform was rated above average and good for the majority of domains.

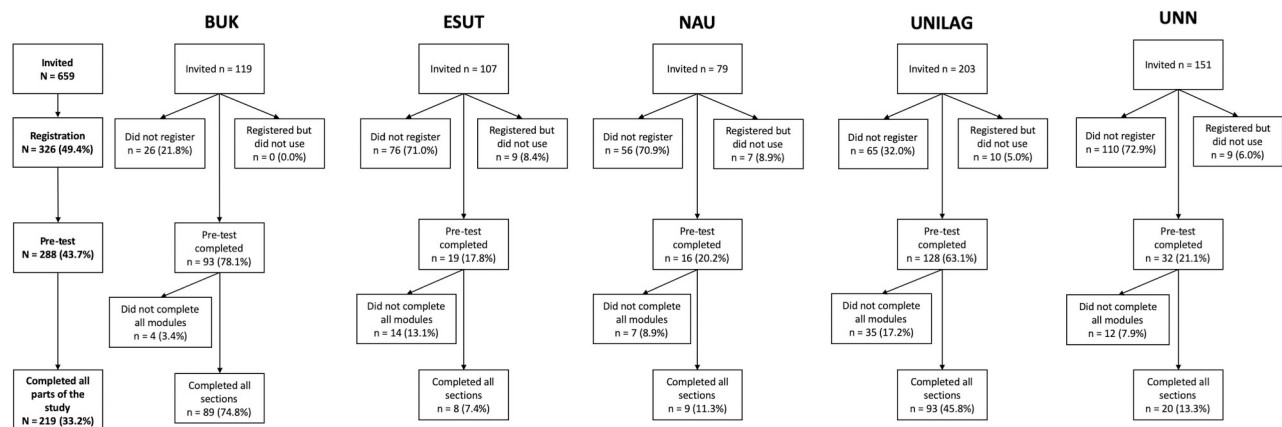

**Fig 2. Summary of participation across different stages of study completion.**

**Table 3. Wilcoxon test to explore effect of module on student's pain management knowledge.**

| Variables | n | Mean (SD) | Median (IQR) | Z-stat | Effect Size (95%CI) | NR PR Ties |
|---|---|---|---|---|---|---|
| Pre-test Overall | 219 | 41.0(13.3) | 40.00(20.00) | -11.3** | 1.3(-0.2 to 2.7) | 19 185 15 |
| Post-test Overall | 219 | 63.2(16.7) | 60.00(25.00) | | | |
| Pre-test BUK | 89 | 40.7(15.7) | 35.00(15.00) | -7.4** | 1.4(-2.5 to 5.4) | 9 75 5 |
| Post-test BUK | 89 | 68.7(20.5) | 65.00(37.50) | | | |
| Pre-test ESUTH | 8 | 38.1(10.0) | 35.00(17.50) | -2.4* | 2.2(0.2 to 4.2) | 0 7 1 |
| Post-test ESUTH | 8 | 60.0(12.0) | 60.00(21.25) | | | |
| Pre-test NAU | 9 | 45.0(13.0) | 50.00(15.00) | -2.4* | 1.5 (-1.1 to 4.0) | 1 7 1 |
| Post-test NAU | 9 | 64.4(15.5) | 60.00(32.50) | | | |
| Pre-test UNILAG | 93 | 41.9(11.8) | 40.00(20.00) | -7.0** | 1.4 (-0.9 to 3.6) | 9 77 7 |
| Post-test UNILAG | 93 | 58.9(12.5) | 60.00(15.00) | | | |
| Pre-test UNN | 20 | 36.8(9.1) | 37.50(13.75) | -3.8** | 2.5(0.9 to 4.1) | 0 19 1 |
| Post-test UNN | 20 | 59.0(8.1) | 60.00(15.00) | | | |

Z = Wilcoxon-Ranked test; BUK = Bayero University, Kano; ESUT = Enugu State University of Science and Technology, Enugu; NAU = Nnamdi Azikiwe University, Awka; UNILAG = University of Lagos; UNN = University of Nigeria Nsukka; PR = Positive Ranks, NR = Negative Ranks;

* p<0.05;

** p<0.001.

Errors logged also highlighted technical reports generated by the e-Learning platform (see Table 6). Technical error logs showed problems with login to be the most frequent issues encountered, with errors also reported relating to downloading and installing the app, and accessing its content. Most error were associated with access to a stable internet connection and using the most up-to-date version of the app.

Analysis of focus group interviews yielded three themes as presented in Table 7: i) Acceptability and engagement with the e-Learning platform; ii) Perceived value of e-Learning approaches for pain; and iii) Recommendations for how engagement and the platform might be developed.

## Discussion

To our knowledge, this is the first study to explore the potential of an online approach to improve the knowledge of pain and its management in medical students studying at colleges of medicine located in an African LMIC. The e-Learning course significantly increased the knowledge of pain management across all five participating sites with student users rated the e-Learning platform positively for both its pragmatic (task-oriented) and hedonic (non-task

**Table 4. Kruskal-Wallis test to compare pre- and post-test scores among schools.**

| Schools | Pre-test Score | | | | Post-test Score | | | |
|---|---|---|---|---|---|---|---|---|
| | Median (IQR) | Median rank test score | $\chi^{2*}$ | p-value | Median (IQR) | Median rank test score | $\chi^{2*}$ | p-value |
| BUK | 35.0(15.0) | 103.5 | 5.3 | 0.26 | 65.0(37.5) | 126.3 | 10.7 | 0.03 |
| ESUT | 35.0(17.5) | 97.6 | | | 60.0(21.3) | 103.6 | | |
| NAU | 50.0(15.0) | 132.2 | | | 60.0(32.5) | 113.5 | | |
| UNILAG | 40.0(20.0) | 118.2 | | | 60.0(15.0) | 97.4 | | |
| UNN | 37.0(13.8) | 95.0 | | | 60.0(15.0) | 97.2 | | |

$\chi^{2*}$ = Kruskal-Wallis test, **BUK** = Bayero University, Kano; **ESUT** = Enugu State University of Science and Technology, Enugu; **NAU** = Nnamdi Azikiwe University, Awka; **UNILAG** = University of Lagos; **UNN** = University of Nigeria, Nsukka.

**Table 5. Median post-test scores in various modules across schools.**

| Module | Median (IQR) | | | | | |
|---|---|---|---|---|---|---|
| | BUK | ESUTH | NAU | UNILAG | UNN | OVERALL |
| 1: Multidimensional nature of pain | 80.0(20.0) | 70.0(20.0) | 60.0(0.0) | 80.0(20.0) | 80.0(0.20) | 80.0(20.0) |
| 2: Neuroanatomy, Neurophysiology and Neuropharmacology of Pain | 80.0(40.0) | 100.0(20.0) | 100.0(20.0) | 80.0(20.0) | 80.0(35.0) | 80.0(20.0) |
| 3: Psychology of pain | 100.0(20.0) | 100.0(20.0) | 80.0(30.0) | 80.0(20.0) | 80.0(20.0) | 80.0(20.0) |
| 4: Pain assessment | 100.0(20.0) | 80.0(0.0) | 80.0(30.0) | 80.0(0.0) | 80.0(35.0) | 80.0(0.0) |
| 5: Treatment of pain | 100.0(20.0) | 80.0(0.0) | 80.0(30.0) | 80.0(0.0) | 80.0(35.0) | 80.0(20.0) |
| 6: Pain in Special situations | 100.0(20.0) | 100.0(0.0) | 100.0(0.0) | 80.0(20.0) | 80.0(20.0) | 100.0(0.20) |

*BUK = Bayero University, Kano; ESUT = Enugu State University Science and Technology, Enugu; NAU = Nnamdi Azikiwe University, Awka; UNILAG = University of Lagos; UNN = University of Nigeria, Nsukka; Module 1 = Multidimensional nature of pain; Module 2 = Neuroanatomy, Neurophysiology and Neuropharmacology of Pain; Module 3 = Psychology of pain; Module 4 = Pain assessment; Module 5 = Treatment of pain; Module 6 = Pain in special conditions.*

oriented) qualities. A lack of capacity within the medical curricula meant the e-Learning platform was provided as an optional activity for students, with uptake from one third of all students who were invited to participate. For those that engaged with the platform, e-Learning was highlighted as a valuable approach to delivering pain education. Factors highlighted as requiring consideration for future development included arranging access points for downloading content at no personal cost to the user, and integration within existing curricula, such as surgical training or internal medicine.

Improved pain management knowledge through e-Learning achieved within this study aligns with previous online programmes evaluated in developed country settings for pain management [21, 22]. Pain beliefs and attitudes are formed during medical education [11]. The e-Learning platform developed within this project provides a resource that could help to address a gap in pain education of undergraduate medical students in Nigeria. This pilot project represented a novel effort to bridge the gap created by the absence of a standardized pain curricula

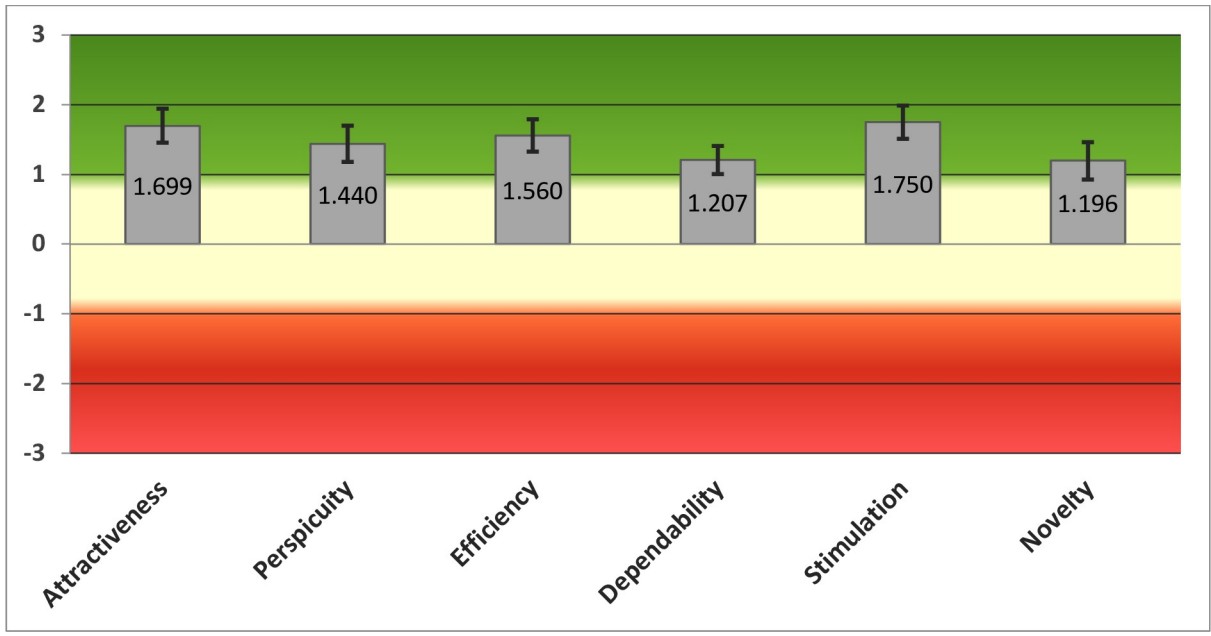

**Fig 3. Mean scores of 6 composite domains with the output generated using the UEQ Data Analysis Tool® version 7.**

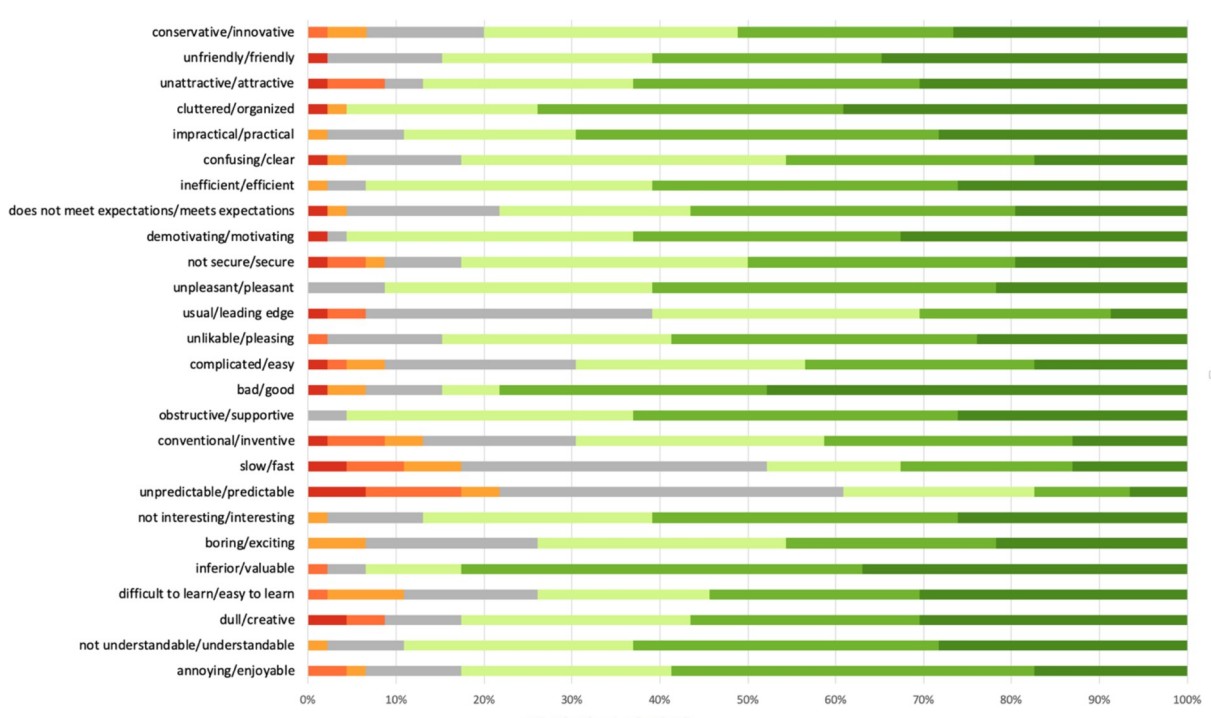

**Fig 4. Distribution of participants' answers across the 7-point scale are presented for each of the 26 items in the user experience questionnaire.** Colour coding varies based on the positive or negative attributes chosen to reflect user perspectives of the e-Learning platform. Colour-coding varied across the 7-point scale from 1 (dark red) suggesting a negative rating against the item (e.g. very conservative or very unfriendly) to 7 (dark green) suggesting a positive rating against the item (e.g. very innovative, very friendly).

in Nigeria. Due to compact curricula across the participating sites, we were not able to integrate the e-Learning platform into ongoing teaching activities. Having demonstrated its ability to improve pain knowledge across participating students, the next crucial phase of development rests on integration within colleges of medicine. This will require an increase in

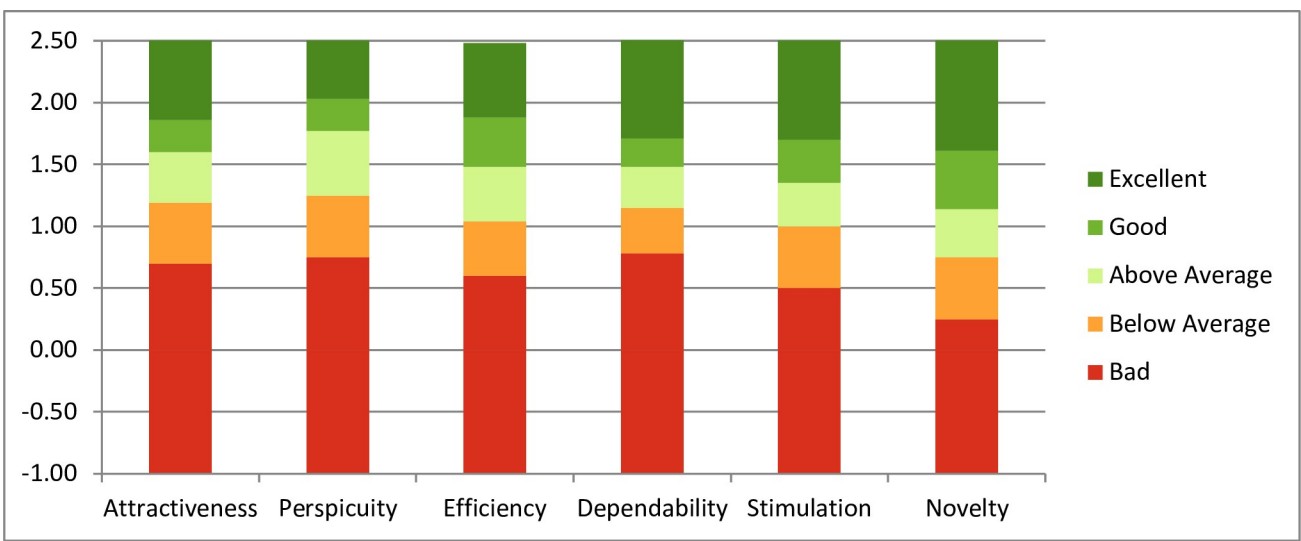

**Fig 5. Quality benchmark graph for Project OPUS training module.**

**Table 6. Overview of IT errors recorded across five sites.**

| Error category | Technical issues | Number of students | Solution |
|---|---|:---:|---|
| **Login** | No login details received | 29 | Sent login details |
| | Login failure due to inputting wrong login ID | 11 | Correct ID shared |
| | login failure due to app issues | 9 | Reinstall latest app version |
| **Download** | Download interrupted due to poor network | 12 | Update app to latest version and redownload with stronger internet connection |
| | Cannot proceed to the next module | 8 | Update app to latest version |
| **Video** | Videos not opening due to app issues | 10 | Update app to latest version |
| | Videos not opening because they were not fully downloaded | 5 | Redownload and wait for the successful download notification |
| **App installation** | Error when updating app | 10 | Sent link to manually update |
| | Unable to install mobile app | 4 | Share link directly via Google Drive or using the web application |

leadership to drive advocacy for adaptation of existing medical school curricula to accommodate a new subspecialty curriculum around pain management. However, e-Learning in medical education is a means to an end, rather than the end in itself [34], and a broader focus on developing the institutional readiness in human and infrastructural resources needed to ensure adoption and sustainability of the platform is now essential. If integration can be achieved, the e-Learning platform could be a tool for health system strengthening to better prepare medical students to support the unmet needs of those with pain in Nigeria and the wider sub-Saharan Africa region [35]. The e-Learning platform developed in this study was supported by a private technology company with expertise in eLearning and an existing track record of supporting health systems strengthening initiatives in Nigeria. The platform has previously been adopted by the government in Nigeria for health worker training in both child and maternal health [36] and disease outbreaks such as Ebola [37]. The next steps of development will require further consideration to determine how to balance access to content and its wider rollout with no or minimal cost to the end user, alongside ensuring alignment to frameworks aiming to enhance the effectiveness of e-learning as an educational tool to increase the quantity and quality of medical education programs [38].

E-Learning offers multiple benefits in the context of developing countries, including flexible learning, time efficiency, reducing costs of printing and paper-based materials, easily modified and updated content, standardization of course content and delivery, the ability to deliver teaching at a distance, and scalability [38]. Despite being optional, a third of all participants completed and sought to engage with the e-Learning platform for pain education, with those who engaged reported many of the beneficial attributes of e-Learning approaches. For those who did not participate, there was a sense of not seeing value, having the time, or the interest in using the platform, alongside experiencing issues with technology; similar experiences have been reported elsewhere [39]. Perceived usefulness of e-Learning can have a strong impact on students' e-Learning intention [40]. The need to advocate for pain education as part of the curriculum is likely to be a crucial component to support further e-Learning approaches, ensuring greater perceived utility of an e-Learning platform that would complement teaching. Furthermore, strategies to improve student engagement in online courses are being developed which may be embedded into future iterations of the platform, including active learning assignments (e.g. discussion boards) which serve to engage students with course content and their peers' course [41].

The study sought to engage students in an online approach to learning but equally adopted an emerging method to capture views and perspectives on the content and structure of the e-

**Table 7. Findings from the Framework Analysis of focus group interviews.**

| Theme | Summary |
|---|---|
| Acceptability and engagement with the e-Learning platform | The e-Learning platform was seen favourably by both those who interacted with the online platform and those that did not. Of those who used and engaged with the platform, it was reported that the platform was useful in "*creating an avenue for me to learn via my phone any day and anytime*" (BUK student 1). The ability to access and complete tasks on the platform was valued across participants. Despite a lack of engagement from some participants, most saw value in increasing teaching and content around pain and felt it would be an informative and useful platform, "*...because it helps us understand the physiology of pain, which is very useful for clinical practice*" (ESUT student 1). |
| | Of those that engaged with the platform, multiple drivers for its use were cited across students. These included those with a "*quest for knowledge*" (ESUT student 2), alongside those wanting to gain knowledge to inform their clinical practice, "*...to be able to do something and save the life of the patient*" (BUK student 2) and act accordingly in the "*...face of an emergency and a patient in pain*" (BUK student 3). |
| | Both barriers and facilitators to engagement were noted for those who did not engage with the platform. For most participants, barriers to use were education-related (e.g. competing deadlines and existing high levels of work across the course), technology-related (e.g. limited phone network signal, lack and expense of data, difficulty downloading the phone application), and personal (e.g. family health, finding the platform exhausting and requiring concentration). A small number of participants were deterred from using the platform due to issues downloading data that were caused by either their mobile phone network coverage or the platform itself. Facilitators to use of the platform noted by participants included individual drive and determination to undertake the training online, perceived cost benefit of online platform when compared to relative expense of buying pain textbooks, and some participants noting offers held by their mobile phone provider that enabled them to allocate more data to accessing and it "*...was helpful in downloading all the section materials*" (ESUT student 3). |
| Perceived value of e-Learning approaches for pain | The appraisal of content by those who engaged with the platform highlighted its importance in recapping on "*...some basics in physiology and biochemistry*" (BUK student 1). However, content was also welcome that taught participants about non-pharmacological approaches to pain management ("*...many more methods of pain management other than drugs*" (ESUT student 1)). Furthermore, participants noted improvements in overcoming confusion around pain management, including that it "*...greatly improved my understanding and cleared multiple misconceptions that are not really taught in a classroom*" (BUK student 4). |
| | In terms of the content covered, there was interest in particular for content covered towards the beginning on the modules, focusing on pain physiology, different types of pain, and definitions surrounding pain and its management. The platform was acknowledged as addressing a broad curriculum, "*ranging from the types and causes of pain to the different approaches in its management*" (ESUT student 4). Furthermore, the content on opioids and end of life care were noted as very important too, which helps to address "*a lot of misconceptions about abuse of opioids. And there's very little end of life pain management provided in this area of the world*" (BUK student 5). |
| | The perception of the platform from peers of those who engaged with the platform indicated that many saw it as an opportunity to acquire new knowledge and it was viewed positively by most peers. The platform was reported as being seen as usable, informative and interesting by peers. However, for some participants, peers reported indifference and lack of interest. For those who did not engage with the platform, some alongside their peers reported that they did not see a need for the platform, viewed it as unnecessary and that the process of completing all activities on the platform took a long time. |
| Recommendations for how engagement and the platform might be developed | Participants who engaged and did not engage with the platform provided multiple suggestions for how the platform and future e-Learning platforms might be implemented in the context of undergraduate medical education in Nigeria. In terms of integration with the existing curriculum, participants felt that it could be included as part of surgical training, as internal medicine, or as "*...a broad aspect of pharmacology or medicine*" (ESUT student 3). Suggestions were provided for how engagement might be increased in future iterations or deployments of the platform. Some participants felt that creating wider awareness of the platform would be advantageous, although many felt that the platform would "*need to be made compulsory*" (UNN student 1) to increase engagement. In terms of platform content, participants suggested that "*reducing the number of video tutorials may make it a bit cheaper as regards data usage*", or that content should be made more concise. Incentives were also suggested by participants, including "*...making the app offline, cutting down on the many modalities*" (ESUT student 4), providing "*...free Wi-Fi strong and accessible by all students*" (LUTH student 1), potentially providing a "*small income because some get discouraged they have to use lots of data without getting paid*" (BUK student 2), or providing participants with more time to access and complete content on the platform. |

Learning platform. Using WhatsApp as a means of conducting focus group interviews is still an evolving research area, providing participants with easier access to interviews, greater freedom to talk about sensitive issues, the ability to express themselves via text and the possibility

of exercising greater control over the interview [42]. Participants were able to freely express positive and negative opinions alongside indifference regarding the platform, whilst providing useful insights into drivers and barriers to engagement. However, as has been highlighted in previous research, WhatsApp group chat may also reduce the quantity and richness of conversation when compared to focus conducted in person [43]. This is an emerging research tool that worked well for undergraduate medical student participants, but may require further innovation and adaptation to evolve alongside phone-based and in-person qualitative research approaches.

This study has limitations. The one-group pre- and post-test design which was used to evaluate changes to pain knowledge has been criticised for its vulnerability to internal validity threats, such as maturation, history, and testing [44]. However, the lack of pain education within the existing curriculum and restricted time between the pre- and post-test measures reduced the impact of possible alternative explanations for the observed differences. Our study limited its focus to pain knowledge, meaning we are unable to determine any influence on skills and attitudes relating to pain management. Participation in the study was voluntary so may have introduced selection bias in the study population, recruiting those most engaged with study. With coverage of one third of all invited participants, this may have led to inflated improvements. However, when considering this at level of site, 74% of those invited at BUK participated, with this site reflecting the largest marked improvement in the post-test as evidenced by mean scores differences. Although, due to low completion rates across some sites the subgroup analyses may have been limited in its ability to detect significant changes in pain knowledge across sites. A further limitation to the study was the budget. This was a pilot study funded through a pump-priming grant. We were unable to support wider costs incurred during delivery, such as data usage for downloading content by students. This may have deterred participation and will be factored in to future iterations, including exploring support that can be provided within institutions to facilitate content downloads. Access to content must be ensured in an equitable way to avoid exacerbating any divide driven by an ability to cover costs of data.

## Conclusion

We outline the development and evaluation of an e-Learning approach to address deficits in existing provision of medical undergraduate training. The e-Learning platform led to improvements in participants' pain knowledge and provided a good user experience; positive outcomes for a target population in which pain beliefs and attitudes are formed. We sought to provide a detailed overview of our development and evaluation process to address criticism of existing research which has failed to provide sufficient detail to support transferability or inform future e-Learning approaches. Future iterations and implementation of the e-Learning platform will be heavily reliant on institutional support, to accommodate pain education within medical school curricula and ensure adequate infrastructure to enable equitable access to content (e.g. facilitating downloading of content). e-Learning has the potential to fundamentally change and shape health systems and the quality of health education. By utilising this approach for pain education in Nigeria, the quality and quantity of health care delivery and access for those with unmet pain needs could be improved.

## Supporting information

**S1 File. Individual-level pain knowledge scores.** Pain knowledge scores attained by students at pre-test, following completion of modules and at post-test.
(PDF)

## Acknowledgments

The authors thank Alexander Kodimalu, Rukayya Umar Yazid, Tomisin Tawose, Fadeel Kabo and Augusta Ezeh for their various roles in student engagement throughout the study period.

Okey Okuzu is Founder and CEO, and Chinelo Oduche a Project Manager, for InStrat Global Health Solutions, the company providing the software on which the e-Learning platform was developed in collaboration with the research team. All other authors declare no conflict of interest.

## Author Contributions

**Conceptualization:** Tonia C. Onyeka, Nneka Iloanusi, Eve Namisango, Matthew J. Allsop.

**Data curation:** Tonia C. Onyeka, Nneka Iloanusi, Eve Namisango, Kehinde S. Okunade, Alhassan Datti Mohammed, Muktar A. Gadanya, Abubakar U. Nagoma, Samuel Ojiakor, Chukwudi Ilo, Okey Okuzu, Chinelo Oduche, Ngozi Ugwu, Matthew J. Allsop.

**Formal analysis:** Tonia C. Onyeka, Nneka Iloanusi, Eve Namisango, Justus U. Onu, Matthew J. Allsop.

**Funding acquisition:** Tonia C. Onyeka, Nneka Iloanusi, Eve Namisango, Matthew J. Allsop.

**Investigation:** Tonia C. Onyeka, Nneka Iloanusi, Eve Namisango, Kehinde S. Okunade, Alhassan Datti Mohammed, Muktar A. Gadanya, Abubakar U. Nagoma, Samuel Ojiakor, Chukwudi Ilo, Matthew J. Allsop.

**Methodology:** Tonia C. Onyeka, Nneka Iloanusi, Eve Namisango, Justus U. Onu, Okey Okuzu, Ngozi Ugwu, Matthew J. Allsop.

**Project administration:** Tonia C. Onyeka, Nneka Iloanusi, Eve Namisango, Kehinde S. Okunade, Alhassan Datti Mohammed, Muktar A. Gadanya, Abubakar U. Nagoma, Samuel Ojiakor, Chukwudi Ilo, Okey Okuzu, Chinelo Oduche, Ngozi Ugwu, Matthew J. Allsop.

**Resources:** Tonia C. Onyeka, Nneka Iloanusi, Eve Namisango, Okey Okuzu, Matthew J. Allsop.

**Software:** Justus U. Onu, Okey Okuzu, Chinelo Oduche, Matthew J. Allsop.

**Supervision:** Matthew J. Allsop.

**Validation:** Tonia C. Onyeka, Nneka Iloanusi, Eve Namisango, Justus U. Onu, Kehinde S. Okunade, Matthew J. Allsop.

**Visualization:** Tonia C. Onyeka, Nneka Iloanusi, Eve Namisango, Justus U. Onu, Matthew J. Allsop.

**Writing – original draft:** Tonia C. Onyeka, Nneka Iloanusi, Eve Namisango, Matthew J. Allsop.

**Writing – review & editing:** Tonia C. Onyeka, Nneka Iloanusi, Eve Namisango, Justus U. Onu, Kehinde S. Okunade, Alhassan Datti Mohammed, Muktar A. Gadanya, Abubakar U. Nagoma, Samuel Ojiakor, Chukwudi Ilo, Okey Okuzu, Chinelo Oduche, Ngozi Ugwu, Matthew J. Allsop.

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
