## [Decision Letter · Decision Letter 0]

11 Sep 2020

PONE-D-20-19766

Project OPUS: Development and evaluation of an electronic platform for pain management education of medical undergraduates in resource-limited settings

PLOS ONE

Dear Dr. Allsop,

Thank you for submitting your manuscript to PLOS ONE. After careful consideration, we feel that it has merit but does not fully meet PLOS ONE’s publication criteria as it currently stands. Therefore, we invite you to submit a revised version of the manuscript that addresses the points raised during the review process.

The recommendation is to revise the manuscript as follows (see for details the enclosed reviewers' comments):

- mention in the limitations of the study that only knowledge acquisition was examined without considering skills and attitude

- show is there is any feedback about comparison of App with other platforms

- clarify if there is any Interactive element in the e-learning system

- clarify if the e-learning system can be used only for self learning or student interaction could be allowed with the teacher and/or with the other students

- use latest IASP 2020 pain definition

- mention other similar courses on pain in developed countries – UK, Europe, Australia, Canada

- discuss implications that VTR Mobile is a proprietary item – or mention under limitations

- correct the citation style per journal requirement

- discuss with more detail differences between completers and non-completers

- write more concisely some areas and summarise the development in a flow diagram

-specify guiding pedagogical theories of online/e-learning that supported the development of the curriculum.

We look forward to receiving your revised manuscript.

Kind regards,

Filomena Papa

Academic Editor

PLOS ONE

Journal Requirements:

"Okey Okuzu is Founder and CEO, and Chinelo Oduche a Project Manager, for InStrat Global Health Solutions, the company providing the software on which the e-Learning platform was developed in collaboration with the research team. All other authors declare no conflict of interest."

We note that one or more of the authors are employed by a commercial company: Instrat Global Health Solutions.

2.1. Please provide an amended Funding Statement declaring this commercial affiliation, as well as a statement regarding the Role of Funders in your study. If the funding organization did not play a role in the study design, data collection and analysis, decision to publish, or preparation of the manuscript and only provided financial support in the form of authors' salaries and/or research materials, please review your statements relating to the author contributions, and ensure you have specifically and accurately indicated the role(s) that these authors had in your study. You can update author roles in the Author Contributions section of the online submission form.

2.2. Please also provide an updated Competing Interests Statement declaring this commercial affiliation along with any other relevant declarations relating to employment, consultancy, patents, products in development, or marketed products, etc.  

Reviewers' comments:

Reviewer's Responses to Questions

**Comments to the Author**

1. Is the manuscript technically sound, and do the data support the conclusions?

Reviewer #1: Yes

Reviewer #2: Yes

2. Has the statistical analysis been performed appropriately and rigorously? 

Reviewer #1: Yes

Reviewer #2: Yes

3. Have the authors made all data underlying the findings in their manuscript fully available?

Reviewer #1: Yes

Reviewer #2: No

4. Is the manuscript presented in an intelligible fashion and written in standard English?

Reviewer #1: Yes

Reviewer #2: Yes

5. Review Comments to the Author

Reviewer #1: This is a very good and much-needed study, with robust methodology and clearly presented findings. However, there are a number of relatively minor comments.

1. Only 1/3rd of the eligible students completed the course – difference between completers and non-completers? Selection bias potential. Has been briefly discussed under limitations, but a comparison would help.

2. Was it at appropriate stage for the students to grasp? What is the medical education system there? It would help the international reader.

3. Knowledge acquisition – what about skills and attitude? Should be mentioned as a limitation.

4. App vs. other platforms? Any feedback?

5. Was there any Interactive element? To what extent?

6. Pain definition has been revised recently – use latest IASP definition 2020.

7. Please mention other similar courses on pain in developed countries – UK, Europe, Australia, Canada.

8. VTR Mobile – proprietary item – concerns about dependence on this particular proprietary item? Needs discussion or mention under limitations.

9. Any reason why pass threshold was set at 70% (Table 1)? Page 13 last sentence says 80%. Please clarify.

10. Table 5: cite only the median with IQR data, not the mean and SD data.

11. Please correct the citation style per Journal requirement (number of authors before et al. can be used).

Reviewer #2: Thank you for submitting this paper for consideration with PLOS one. This is an interesting study that was funded by a leading international body, the International Association for the Study of Pain. The development and evaluative research are carefully described and the study uses mixed and multi-methods to evaluate a pain curriculum and mobile platform for use in Nigeria. Some areas of the paper could be more concisely written which is difficult with a complex project and data. Some specific comments are below that I hope are helpful in refining the paper.

A structured abstract that provides a good summary of the project

Introduction

A well written and interesting introduction that provides a strong rationale for the project.

First line describes pain as a very frequent complaint. Please consider revising this as representing pain as a complaint suggests that it is minor, a symptom rather than a complex condition or experience that can have a major impact on people. Also, this emphasizes the reporting of pain and there are many who cannot express pain or are reluctant to do so.

IASP definition of pain has recently been updated and needs to be reflected in the paper

Overview of platform development – HIT acronym needs explaining/writing out

This paper describes the detailed development of the platform and curriculum which is one of its strengths. Anyone developing something similar would find this information helpful. However, some areas could be more concisely written and the development could be summarised in a flow diagram to reduce the word count.

As well as the review to determine the pain-related content, were there any guiding pedagogical theories of online/e-learning that supported the development of the curriculum?

Research tools: Knowledge survey –what was covered in this? Pain Beliefs Survey – reference and details needed. Whether reliability and validity testing was conducted on these is needed

Page 15 - In this sentence, change done to assessed ‘The normality of distribution of the data was done…’ and later in the same paragraph – undertaken is another option

Results section

Page 16, third line- should this be completion rate rather than retention rate? And again two lines later. Not sure retention is the correct term

Table 3 states that 8 people completed the pre and post test from the ESUTH site but in Figure 2, this number is 5. Please can this be corrected with whichever is correct.

Table 7 presents the qualitative analysis but it is unclear why a table has been used to present nearly 900 words of text. Consider revising.

Discussion

A strong discussion. Some acknowledgement of the limitations around the smaller sub-groups (sites) and statistical analysis would be important

Conclusion

First line needs revision as it raises much broader issues that are not the focus of this paper, particularly opioid abuse/misuse. The line reads ‘Inadequate pain education will perpetuate existing public health problems of undertreated, pain, untreated pain and opioid abuse

Thank you for the opportunity to review the paper and good luck with the next stages of development for the project.

6. PLOS authors have the option to publish the peer review history of their article (what does this mean?). If published, this will include your full peer review and any attached files.

Reviewer #1: **Yes: **Sukanya Mitra

Reviewer #2: **Yes: **Dr Emma Briggs

---

## [Author Response · Author response to Decision Letter 0]

21 Sep 2020

Please see attached our cover letter and 'Response to Reviewers' document outlining our point-by-point response.

---

## [Decision Letter · Decision Letter 1]

11 Nov 2020

PONE-D-20-19766R1

Project OPUS: Development and evaluation of an electronic platform for pain management education of medical undergraduates in resource-limited settings

PLOS ONE

Dear Dr. Allsop,

Thank you for submitting your manuscript to PLOS ONE. After careful consideration, we feel that it has merit but does not fully meet PLOS ONE’s publication criteria as it currently stands. Therefore, we invite you to submit a revised version of the manuscript that addresses the points raised during the review process.

It is recommended to carefully revise the manuscript including modifications suggested by reviewer 3, correcting typos (e. g. in section data analysis "pre-and post-test", page 8 "pre and post-test", "user-centered design[25]", "retirement.[24]") and eliminating discrepancies in notation (e .g. page 18 "Pre and post test scores " and "The pre-test, post-test...", Table 5 "Median Posttest").

We look forward to receiving your revised manuscript.

Kind regards,

Filomena Papa

Academic Editor

PLOS ONE

Reviewers' comments:

Reviewer's Responses to Questions

**Comments to the Author**

1. If the authors have adequately addressed your comments raised in a previous round of review and you feel that this manuscript is now acceptable for publication, you may indicate that here to bypass the “Comments to the Author” section, enter your conflict of interest statement in the “Confidential to Editor” section, and submit your "Accept" recommendation.

Reviewer #1: All comments have been addressed

Reviewer #3: (No Response)

2. Is the manuscript technically sound, and do the data support the conclusions?

Reviewer #1: Yes

Reviewer #3: Yes

3. Has the statistical analysis been performed appropriately and rigorously? 

Reviewer #1: Yes

Reviewer #3: Yes

4. Have the authors made all data underlying the findings in their manuscript fully available?

Reviewer #1: Yes

Reviewer #3: Yes

5. Is the manuscript presented in an intelligible fashion and written in standard English?

Reviewer #1: Yes

Reviewer #3: Yes

6. Review Comments to the Author

Reviewer #1: All the comments have been responded to my satisfaction. Corrections and changes have been made where possible and needed, and mentioned as limitations where needed but not possible. This version appears acceptable to me now. Congratulations on a very important work!

Reviewer #3: This original research paper evaluated the implementation of a voluntarily 12-week e-learning course on pain education across five medical schools in Nigeria, with positive outcomes in regards to implementation of the method of e-learning and increased knowledge on pain. The advantages and limitations of this study have been nicely discussed. I want to congratulate the authors on their nice work on an important topic.

I have minor suggestions for revision:

1. The whole reference list needs to be revised due to errors and mixups. As an example, see refs 28 and 33.

2. In the abstract, result section, perhaps it would be nice to state the collective pre- and post test scores across all medical schools in numbers and not only mention the good effect sizes. If word count is an issue I suggest removing (for instance) the mention of what statistical test used to compare pre- and post scores.

3. Methods. Very precisely described in regards to the platform. However, I would like some more information on how the selection of different sections from the IASP curriculum was made. Information on why the whole curriculum was not used and why additional resources were used to form the teaching material for this course. I think you should also add a reference to the specific curriculum outline used on the IASP website (if any).

4. On page 14, Method section; it is stated that the students were able to retake the tests. I recommend stating if this relates to the tests taken after completing each section (or if it also includes the pre/post tests.)

5. The pain beliefs survey mentioned in the methods; as far as I can see these results are not presented in this paper and the reference to its validation is not made on the students from this study, so why is this mentioned in the methods? Please clarify.

7. PLOS authors have the option to publish the peer review history of their article (what does this mean?). If published, this will include your full peer review and any attached files.

Reviewer #1: **Yes: **Sukanya Mitra

Reviewer #3: **Yes: **Linda Rankin

---

## [Author Response · Author response to Decision Letter 1]

17 Nov 2020

Please see attached our cover letter and 'Response to Reviewers' document outlining our point-by-point response.

---

## [Decision Letter · Decision Letter 2]

24 Nov 2020

Project OPUS: Development and evaluation of an electronic platform for pain management education of medical undergraduates in resource-limited settings

PONE-D-20-19766R2

Dear Dr. Allsop,

We’re pleased to inform you that your manuscript has been judged scientifically suitable for publication and will be formally accepted for publication once it meets all outstanding technical requirements.

Kind regards,

Filomena Papa

Academic Editor

PLOS ONE

Additional Editor Comments (optional):

Reviewers' comments:

Reviewer's Responses to Questions

**Comments to the Author**

1. If the authors have adequately addressed your comments raised in a previous round of review and you feel that this manuscript is now acceptable for publication, you may indicate that here to bypass the “Comments to the Author” section, enter your conflict of interest statement in the “Confidential to Editor” section, and submit your "Accept" recommendation.

Reviewer #1: All comments have been addressed

Reviewer #3: All comments have been addressed

2. Is the manuscript technically sound, and do the data support the conclusions?

Reviewer #1: Yes

Reviewer #3: Yes

3. Has the statistical analysis been performed appropriately and rigorously? 

Reviewer #1: Yes

Reviewer #3: Yes

4. Have the authors made all data underlying the findings in their manuscript fully available?

Reviewer #1: Yes

Reviewer #3: Yes

5. Is the manuscript presented in an intelligible fashion and written in standard English?

Reviewer #1: Yes

Reviewer #3: Yes

6. Review Comments to the Author

Reviewer #1: No further comments. All the comments have been responded to

my satisfaction. Corrections and changes

have been made where possible and needed,

and mentioned as limitations where needed

but not possible. This version appears

acceptable to me now. Congratulations on a

very important work!

Reviewer #3: All my comments and recommendations have been met by the authors. I therefore recommend for this paper to be accepted.

7. PLOS authors have the option to publish the peer review history of their article (what does this mean?). If published, this will include your full peer review and any attached files.

Reviewer #1: **Yes: **Prof. Sukanya Mitra

Reviewer #3: **Yes: **Linda Rankin

---

## [Editor Report · Acceptance letter]

26 Nov 2020

PONE-D-20-19766R2 

Project OPUS: Development and evaluation of an electronic platform for pain management education of medical undergraduates in resource-limited settings 

Dear Dr. Allsop:

I'm pleased to inform you that your manuscript has been deemed suitable for publication in PLOS ONE. Congratulations! Your manuscript is now with our production department. 

Kind regards, 

on behalf of

Dr. Filomena Papa 

Academic Editor

PLOS ONE